# Morbidities, health problems, health care seeking and utilization behaviour among elderly residing on urban areas of eastern Nepal: A cross-sectional study

Mukesh Poudel[1]*, Asmita Ojha[2], Jeevan Thapa[3], Deepak Kumar Yadav[4], Ram Bilakshan Sah[4], Avaniendra Chakravartty[4], Anup Ghimire[4], Shyam Sundar Budhathoki[5,6]

1 Epidemiology and Disease Control Division, Ministry of Health and Population, Kathmandu, Nepal, 2 Health Office, Nuwakot, Ministry of health, Bagmati Province, Hetauda, Nepal, 3 Department of Community Health Sciences, Patan Academy of Health Sciences, Lalitpur, Nepal, 4 School of Public Health & Community Medicine B.P Koirala Institute of Health Sciences, Dharan, Nepal, 5 Department of Primary Care and Public Health, School of Public Health, Imperial College London, London, United Kingdom, 6 Golden Community, Lalitpur, Nepal

* poudelmuk@gmail.com

**Data Availability Statement:** Uploaded as supporting information.

## Abstract

### Background

Morbidity increases with age and enhances the burden of health problems that result in new challenges to meet additional demands. In the ageing population, health problems, and health care utilization should be assessed carefully and addressed. This study aimed to identify chronic morbidities, health problems, health care seeking behaviour and health care utilization among the elderly.

### Methods

We conducted a community based, cross-sectional study in urban areas of the Sunsari district using face-to-face interviews. A total of 530 elderly participants were interviewed and selected by a simple proportionate random sampling technique.

### Results

About half, 48.3%, elderly were suffering from pre-existing chronic morbidities, of which, 30.9% had single morbidity, and 17.4% had multi-morbidities. This study unfurled more than 50.0% prevalence of health ailments like circulatory, digestive, eye, musculoskeletal and psychological problems each representing the burden of 68.7%, 68.3%, 66.2%, 65.8% and 55.7% respectively. Our study also found that 58.7% preferred hospitals as their first contact facility. Despite the preferences, 46.0% reported visiting traditional healers for treatment of their ailments. About 68.1% reported having difficulty seeking health care and 51.1% reported visits to a health care facility within the last 6 months period. The participants with pre-existing morbidity, health insurance, and an economic status above the poverty line were more likely to visit health care facilities.

**Funding:** The author(s) received no specific funding for this work.

**Competing interests:** The authors have declared that no competing interests exist.

## Conclusion

Elderly people had a higher prevalence of health ailments, but unsatisfactory health care seeking and health care utilization behaviour. These need further investigation and attention by the public health system in order to provide appropriate curative and preventive health care to the elderly. There is an urgent need to promote geriatric health services and make them available at the primary health care level, the first level of contact with a national health system.

## Introduction

Ageing is an inevitable biological process that renders physical, psychological, and social transformation with an increasing possibility of complex and expensive diseases [1]. The global ageing population is on the increasing trend, hence to fulfil the pledge of the 2030 agenda, "leave no one behind," it is essential to prepare for an effective way to deal with the contextual health needs of the ageing population [2].

Morbidity increases with age, thus health care systems need to continually update in order to adequately cater for these additional demands [3]. Studies report the existence of inequalities in health services with sustained unmet needs for healthcare among the elderly population [4–6]. Lower survival probabilities for individuals with unmet health care needs are reported. Moreover, unmet health needs conferred a higher risk of mortality, with excess risk ranging from 10% to 155% [7].

Achieving healthy ageing relates to the adequate maintenance of functional ability by appropriately addressing the health care needs of the elderly [8]. Health needs assessments typically rely on the measurement of the health status and assessment of the services that are required in a community to highlight the key problems. Needs assessment enables the identification of the unmet needs to address the gaps in receipt of assistance for healthful ageing [9].

In Nepal, the elderly refers to a senior citizen who has completed the age of sixty years [10]. The elderly population of Nepal is 8.1% in 2011 and is in an increasing trend with the growth rate of the ageing population around 3.5% [11, 12]. The Global Age Watch Index of Help age International 2015 proclaimed Switzerland as the most suitable country for the old to live whereas Nepal ranks 70 among 96 in the index [13]. The increase in the ageing population and the sustained shift in population age structure pose an array of challenges to overall health services and policymakers to combat different morbidity patterns among the elderly residing in Nepal [14]. There is a lack of awareness in addressing the concerns of the ageing population in Nepal. Social protection is limited and most of the needs of the elderly are unmet with a lack of sensitivity and awareness regarding the rights of the elderly [12].

The assessment of the chronic morbidities, health problems among the elderly of Nepal, their health care seeking and utilization behaviour prevalent in society will be helpful to plan relevant interventions focusing on healthy ageing. This study aims to identify the preexisting chronic morbidities, health problems, health care seeking and utilization behaviours among the elderly population.

## Materials and methods

### Study design

This was a community-based cross-sectional study involving a face-to-face interview with elderly people residing in two metropolitan cities of Sunsari district during 2017 and 2018.

## Study setting

**General setting.**    Nepal is a landlocked country in South Asia that borders China in the north and India in the east, west and south. It is a federal republic comprising seven provinces with a population of 30.2 million as per the ongoing census 2021 [15]. The elderly people constitute 8.14% with an increase in life expectancy to 68 years in 2012. The total population growth rate of Nepal is about 1.4% while the growth rate of the ageing population is around 3.5% [11, 12].

**Specific setting.**    Among the seven provinces of Nepal, Province one is the easternmost part. It consists of 14 districts, one metropolitan city, two sub-metropolitan cities, 46 municipalities and 88 rural municipalities. Itahari and Dharan are the second and third largest cities and two sub-metropolitan cities of Province one located in Sunsari District [16]. Both consist of 20 wards each [17, 18]. We conducted the study from September 2017 to August 2018 in randomly selected 16 wards of Itahari and Dharan (eight wards from each city).

## Participants

We selected the household by stratified proportionate random sampling method. The two sub-metropolitan cities of Sunsari district were considered urban areas and constituted the sampling frame for our study. Eight wards selected randomly from each sub-metropolitan city were considered strata. The ward-wise population of the sub-metropolitan city was collected from the administrative office and the number of participants from each ward was calculated proportionately [17, 18]. Subjects above the age group of 60 years and older and willing to participate in the study were included following approval of detailed informed consent. We took a bottle and rotated it at a centrally located spot in each ward. The first household was the direction pointed by the bottle. Thereafter, we visited every third house until the desired sample size was fulfilled. In case of unavailability of the sample population in selected households, the adjacent household was considered for data collection. If more than one member from the same household met the sample criteria, one member was chosen by the lottery method.

The sample size was calculated at a 95% confidence interval and 85% power based on one proportion sample size formula $n = Z^2PQ/D^2$. The prevalence of unmet health care needs in people suffering from hypertension was 26.2% in a study done in Bhaktapur District [3]. This prevalence suggests the scenario of health care seeking practice with one of the common morbidity among the elderly. The value of Z is 1.96 at a 95% confidence interval, p was 26.2, q was 73.8(100-p) and d is 15% of p at 85% power. The total calculated sample size after adding a 10% to adjust for potential non-responders was 530.

## Variables studied

We did a face-to-face interview using a semi-structured pre-tested questionnaire to gather information regarding demographics and other variables. A questionnaire intended to fulfil the study objective was developed based on different studies among the elderly. The questions were discussed among authors for validity. The translation of questions into the local language was done. The elderly visiting the BP Koirala Institute of Health Sciences was interviewed. The authors reviewed the acquired answers with an amendment to the questionnaire. Participants identified their pre-existing chronic morbidity from a list of five chronic diseases used by the package for essential non-communicable diseases in Nepal [19].

Participants went through a series of questions about their ability to see newspaper print, the ability to see the face of someone four-meter away clearly, the ability to hear clearly in a conversation with one other person, and the ability to chew hard foods without difficulty to identify the problems in physical and sensory functions [4].

Moreover, we asked the participants if they had any additional health problems from a list of common presenting symptoms in the elderly identified by a study in Chandigarh [20].

Blood pressure was measured by the interviewer using a calibrated aneroid sphygmomanometer on the right arm with the participant sitting on a chair with their arm resting on the table at heart level, using an appropriately sized cuff [21]. Hypertension was classified according to the seventh report of the joint national committee in not previously diagnosed cases [22].

We assessed depression using the BDI-II scale translated into Nepali which is a validated tool for use in Nepal [23]. In non-clinical populations, scores above 20 indicate depression. The Cronbach's alpha of the BDI-II scale has shown good consistency of 0.76 [24]. Depression is one of the most common psychological problems in the elderly population [25]. The findings are presented as a psychological problem in the result section.

The interviewer performed a general physical examination to assess pulse rate and identify pallor, icterus, lymphadenopathy, cyanosis, clubbing, oedema and dehydration.

The complaints and findings were then stratified into various health problems according to the International classification for primary care (ICPC) [26].

Ethnicity was categorized per the Health Management Information System (HMIS) based on the central bureau of statistics population monograph of Nepal [27].

The poverty line was according to the world bank global poverty line revised in October 2015 (US$1.90 per day) [28]. The exchange rate used is the one fixed by Nepal Rastra bank at the time of analysis on 1st October 2018. USD 1$ = 115 NRs [29].

Current tobacco users are those using any form of tobacco daily or had used it in the past 30 days. Past tobacco users are those who said they had used tobacco once for a few months or years in the past [21].

Physical activity was termed as an adequate amount if the participant does at least 150 minutes of moderate-intensity aerobic physical activity in one week or does at least 75 minutes of vigorous-intensity aerobic physical activity throughout the week or an equivalent combination of moderate and vigorous-intensity activity [30].

Fruit and vegetable intake was said to be adequate if the participant consumes more than five servings of fruit and vegetable in one day [21].

Self-reported health status is the self-rating of the participant's health status in terms of good, average, and poor.

## Statistical analysis

Statistical analysis was done using the statistical package for social sciences (SPSS version 16). Data are described using frequency and percentage for categorical data, while continuous data are presented in mean and standard deviation. The association of independent variables with difficulty in health care seeking and health care utilization was assessed using the chi-square test in bivariate analysis. Multivariable analysis was done using conditional logistic regression [backward] to calculate the adjusted odds ratio of the independent variables. All the variables with a p-value less than 0.25 were considered for multivariable analysis and were tested for collinearity (to include those with VIF < 2). All the analysis was done at a 5% significance level considering a p-value of less than 0.05 as significant.

## Ethical clearance

The Institutional Review Committee of B.P Koirala Institute of Health Sciences (Code No. IRC/1163/017) provided the approval for the study. The permission to conduct the study in the communities was obtained from the respective metropolitan office.

## Results

### Socio-demographic and behavioural characteristics

This study constituted 49.1% of females. The mean age of the participant in this study was 72.2 years with a standard deviation of 8.1 years. The participants were in-between ages ranging from 60 years to 101 years. The highest proportion of participants belonged to the age group 60–69 years (41.1%), Brahmin/Chettri ethnicity (45.6%) and Hindu religion (85.7%). Among the study participants, 63.8% were currently living as a couple, and 87.0% of the elderly were living with their families. More than half (56.0%) of the participants were below the poverty line. More than two-thirds of the participants in our study (67.7%) were not able to read and write. In our study, 36.8% and 35.5% reported current tobacco and alcohol use respectively. Approximately, a quarter (26.8%) reported adequate consumption of fruit and vegetable, and 60.0% performed adequate physical activity (Table 1).

Table 1. Sociodemographic and behavioural characteristics of participants [n = 530].

| Characteristic | Categories | Frequency (n) | Percentage (%) |
|---|---|---|---|
| Age Distribution | 60–69 years | 218 | 41.1 |
| | 70–79 years | 209 | 39.5 |
| | 80 years and above | 103 | 19.4 |
| | Mean ± SD (min, max) | 72.2 ± 8.1 (60, 101) | |
| Gender | Male | 270 | 50.9 |
| | Female | 260 | 49.1 |
| Ethnicity | Dalit | 53 | 10.0 |
| | Janajati | 224 | 42.3 |
| | Madhesi/ Muslim | 11 | 2.1 |
| | Brahmin/Chettri | 242 | 45.6 |
| Relationship status | Single* | 192 | 36.2 |
| | Couple | 338 | 63.8 |
| Living arrangement # | With family | 461 | 87.0 |
| | By themselves | 69 | 13.0 |
| Economic status | Below Poverty Line | 297 | 56.0 |
| | Above Poverty line | 233 | 44.0 |
| Ability to read and write | No | 359 | 67.7 |
| | Yes | 171 | 32.3 |
| Tobacco use | Current user | 195 | 36.8 |
| | Past user | 199 | 37.5 |
| | Never user | 136 | 25.7 |
| Alcohol Use | Current Drinker | 188 | 35.5 |
| | Past 12 months abstainer | 107 | 20.2 |
| | Lifetime abstainer | 235 | 44.3 |
| Fruit and veg intake | Adequate | 142 | 26.8 |
| | Inadequate | 388 | 73.2 |
| Physical Activity | Low activity | 212 | 40.0 |
| | Adequate activity | 318 | 60.0 |
| Self-reported health status | Good | 92 | 17.4 |
| | Average | 275 | 51.9 |
| | Poor | 163 | 30.8 |

* Single constituted 3 never married, 185 widowed, 2 separated and 2 divorced participants

# Participants living with their son or daughter were considered living with family and those living alone or with a spouse were considered living by themselves.

**Table 2. Self-reported pre-existing chronic morbidities and health problems among the elderly [n = 530].**

| Characteristics | Categories | Frequency | Percentage |
| --- | --- | --- | --- |
| Pre-existing Morbidities | None | 274 | 51.7 |
| | Single condition | 164 | 30.9 |
| | Multimorbidity | 92 | 17.4 |
| Chronic Morbidities | Hypertension | 180 | 34.0 |
| | Diabetes | 76 | 14.3 |
| | Cardiovascular Disease | 57 | 10.8 |
| | COPD/Asthma | 52 | 9.8 |
| | Cancer | 4 | 0.8 |
| Health Problems | Circulatory problem | 367 | 69.3 |
| | Digestive problem | 362 | 68.3 |
| | Eye problem | 351 | 66.2 |
| | Musculoskeletal problem | 349 | 65.8 |
| | Psychological problem | 295 | 55.7 |
| | Urological problem | 164 | 30.9 |
| | Ear problem | 158 | 29.8 |
| | Skin problem | 128 | 24.2 |
| | Respiratory problem | 111 | 20.9 |
| | Neurological problem | 110 | 20.8 |
| | General unspecified problem | 94 | 17.7 |
| | Endocrine / metabolic problem | 76 | 14.3 |
| | Genital problem | 31 | 5.8 |

Health problems are classified according to the complaints/ symptoms and examination findings as per the International Classification of primary care [ICPC-2]

Our study identified that about half (48.3% elderly were suffering from pre-existing chronic morbidities, of which, 30.9% had single morbidity, and 17.4% had multimorbidity. Hypertension was the most common morbidity reported by 34.0% followed by diabetes mellitus among 14.3%. Hypertension with diabetes mellitus was the commonest multimorbidity reported by 9.4% of respondents. Almost, all the participants except three reported having health needs for different ailments at the time of the interview, of which problem relating to the circulatory system was the commonest (68.7%) followed by digestive problems (68.3%). Moreover, the prevalence of problems with the eye, musculoskeletal and psychological was more than 50 per cent (Table 2).

## Health care seeking and utilization behaviour

In this study, 19.2% reported difficulty in performing daily activities and 87.0% were able to leave home without help. Approximately half, 51.9% of the participants reported having average health status. The majority of the participants (58.7%) preferred to visit a hospital in the case of need while 6.2% reported a preference for traditional healers and 5.1% hesitated to go anywhere confining themselves in their houses. Despite the lower preference for first contact, almost half of the participants (46.0%) reported visiting traditional healers for their illnesses. Table 3 lists various health care seeking behaviour and health service utilization characteristics among the elderly.

Table 4 list the association of different variables with difficulty in seeking health care and health care utilization within six months. The studied variables like age, current occupation, economic status, ability to read and write, self-reported health status, health insurance and

**Table 3. Health care seeking and utilization behaviour among the participants [n = 530].**

| Characteristic | Category | Total [n] | Percentage [%] |
|---|---|---|---|
| **Preferred health facility for first contact** | Hospital | 311 | 58.7 |
| | Pharmacy | 55 | 10.4 |
| | Private Clinic | 51 | 9.6 |
| | Traditional healer | 33 | 6.2 |
| | PHC/Health post | 30 | 5.7 |
| | Pension camp* | 23 | 4.3 |
| | Nowhere/Home Remedy | 27 | 5.1 |
| **Walking distance to the nearest health facility** | 30 minute or less | 345 | 65.1 |
| | More than 30 minute | 185 | 34.9 |
| **Last visit to any health facility** | Less than six months | 271 | 51.1 |
| | Six to less than 12 months | 56 | 10.6 |
| | one to three years | 57 | 10.8 |
| | More than three years | 99 | 18.7 |
| | Never | 47 | 8.9 |
| **Number of health facility visits in last six months** | None | 257 | 48.5 |
| | Once | 112 | 21.1 |
| | Twice | 93 | 17.5 |
| | Thrice or more | 68 | 12.8 |
| **Sought emergency care in last six months** | Yes | 55 | 10.4 |
| | No | 475 | 89.6 |
| **Admitted for inpatient care in last six months** | Yes | 50 | 9.4 |
| | No | 480 | 90.6 |
| **Health insurance** | Insured | 114 | 21.5 |
| | Not Insured | 416 | 78.5 |
| **Difficulty in seeking health care** | Yes | 361 | 68.1 |
| | No | 169 | 31.9 |
| **Perception about health services** | Satisfactory | 382 | 72.1 |
| | Non-satisfactory | 148 | 27.9 |
| **Awareness of the government geriatric health scheme** | Yes | 256 | 48.3 |
| | No | 274 | 51.7 |

* Pension camp is the common term used by people for pension paying office for the Ex Gurkha army officials, which also have a health facility with medical personnel and doctor which provides health care service free of cost.

awareness of the government geriatric health scheme were significant contributors to the difficulties in seeking health care, and health care utilization among the elderly. Moreover, gender, relationship status and walking distance of the nearest health facility had a measurable impact on difficulty seeking health care while preexisting morbidities were related to health care utilization (Table 4).

Tables 5 and 6 show the results of multivariable analyses of different variables associated with difficulty in health-seeking care and health care utilization respectively. In our study, female gender, those below the poverty line, those with average to poor self-reported health status, non-availability of the nearest health facility within 30 minutes of walking distance and those who were not aware of government geriatric health schemes significantly reported difficulty in seeking health care. The elderly with a poor self-reported health status were approximately 9.6 times more likely to face difficulty in seeking health care. (Table 5) Expectedly, elderly of economic status above the poverty line, insured for health and those who were aware

**Table 4. Bivariate analysis of different variables with difficulty in seeking health care and health care utilization within six months.**

| | Difficulty in seeking health care | | | Health care utilization within six months | | |
|---|---|---|---|---|---|---|
| | Yes 361(68.1%) | No 169(31.9%) | p-value | Yes 271(51.1%) | No 259(48.9%) | p-value |
| **Age in years** | | | | | | |
| Less than 70 | 134[61.5%] | 84[38.5%] | 0.01 | 104[47.7%] | 114[52.3%] | 0.04 |
| 70 to 79 | 147[70.3%] | 62[29.7%] | | 121[57.9%] | 88[42.1%] | |
| 80 and above | 80[77.7%] | 23[22.3%] | | 46[44.7%] | 57[55.3%] | |
| **Gender** | | | | | | |
| Male | 151[55.9%] | 119[44.1%] | 0.00 | 133[49.3%] | 137[50.7%] | 0.4 |
| Female | 210[80.8%] | 50[19.2%] | | 138[53.1%] | 122[46.9%] | |
| **Living arrangement** | | | | | | |
| Living with family | 312[67.7%] | 149[32.3%] | 0.60 | 243[52.7%] | 218[47.3%] | 0.06 |
| Living by themselves | 49[71.0%] | 20[29.0%] | | 28[40.6%] | 41[59.4%] | |
| **Relationship status** | | | | | | |
| Couple | 210[62.1%] | 128[37.9%] | 0.00 | 173[51.2%] | 165[48.8%] | 0.98 |
| Single | 151[78.6%] | 41[21.4%] | | 98[51.0%] | 94[49.0%] | |
| **Current occupation** | | | | | | |
| Paid Job | 11[50.0%] | 11[50.0%] | 0.00 | 7[31.8%] | 15[68.2%] | 0.03 |
| Agriculture | 23[51.1%] | 22[48.9%] | | 16[35.6%] | 29[64.4%] | |
| Business | 23[48.9%] | 24[51.1%] | | 25[53.2%] | 22[46.8%] | |
| Homemaker/Retired | 304[73.1%] | 112[26.9%] | | 223[53.6%] | 193[46.4%] | |
| **Economic status** | | | | | | |
| Below Poverty Line | 223[75.1%] | 74[24.9%] | 0.00 | 120[40.4%] | 177[59.6%] | <0.01 |
| Above Poverty line | 138[59.2%] | 95[40.8%] | | 151[64.8%] | 82[35.2%] | |
| **Ability to read and write** | | | | | | |
| Yes | 89[52.0%] | 82[48.0%] | 0.00 | 100[58.5%] | 71[41.5%] | 0.02 |
| No | 272[75.8%] | 87[24.2%] | | 171[47.6%] | 188[52.4%] | |
| **Self-reported health status** | | | | | | |
| Good | 34[37.0%] | 58[63.0%] | 0.00 | 39[42.4%] | 53[57.6%] | 0.2 |
| Moderate | 183[66.5%] | 92[33.5%] | | 145[52.7%] | 130[47.3%] | |
| Poor | 144[88.3%] | 19[11.7%] | | 87[53.4%] | 76[46.6%] | |
| **Pre-existing morbidity** | | | | | | |
| None | 186[67.9%] | 88[32.1%] | 0.15 | 97[35.4%] | 177[64.6%] | <0.01 |
| Single Condition | 119[72.6%] | 45[27.4%] | | 99[60.4%] | 65[39.6%] | |
| Multimorbidity | 56[60.9%] | 36[39.1%] | | 75[81.5%] | 17[18.5%] | |
| **Health Insurance** | | | | | | |
| Yes | 66[57.9%] | 48[42.1%] | 0.01 | 86[75.4%] | 28[24.6%] | <0.01 |
| No | 295[70.9%] | 121[29.1%] | | 185[44.5%] | 231[55.5%] | |
| **Walking distance to the nearest health facility** | | | | | | |
| 30 minutes or less | 213[61.7%] | 132[38.3%] | 0.00 | 173[50.1%] | 172[49.9%] | 0.5 |
| More than 30 minutes | 148[80.0%] | 37[20.0%] | | 98[53.0%] | 87[47.0%] | |
| **Awareness of the government geriatric health scheme** | | | | | | |
| Yes | 151[59.0%] | 105[41.0%] | 0.00 | 173[67.6%] | 83[32.4%] | <0.01 |
| No | 210[76.6%] | 64[23.4%] | | 98[35.8%] | 176[64.2%] | |

of government geriatric health schemes were more likely to visit health care facilities within six months. Moreover, among the elderly with pre-existing morbidity, those with single morbidity and multimorbidity were at 2.4 and 6.2 higher odds of health care utilization respectively (Table 6).

**Table 5. Multivariable logistic regression analysis of different variables with difficulty in health-seeking behaviour.**

| Characteristic | Categories | AOR | 95% C.I.for Adjusted Odds Ratio (AOR) | | p-value |
|---|---|---|---|---|---|
| | | | Lower | Upper | |
| **Gender** | Male | REF | | | <0.01 |
| | Female | 2.492 | 1.544 | 4.022 | |
| **Relationship status** | Couple | REF | | | 0.09 |
| | Single | 1.536 | 0.929 | 2.54 | |
| **Economic Status** | Above poverty line | REF | | | 0.01 |
| | Below Poverty line | 1.751 | 1.13 | 2.712 | |
| **Self-reported health status** | Good | REF | | | <0.01 |
| | Average | 3.585 | 2.076 | 6.193 | |
| | Poor | 9.86 | 4.92 | 19.759 | |
| **Walking distance to the nearest health facility** | 30 minutes or less | REF | | | <0.01 |
| | More than 30 minutes | 3.609 | 2.182 | 5.971 | |
| **Aware of the government geriatric health scheme** | Yes | REF | | | <0.01 |
| | No | 3.161 | 2.002 | 4.989 | |

Variables adjusted with age, gender, relationship Status, occupation, economic status, ability to read and write, self-reported health status, preexisting morbidity, health insurance, walking distance to the nearest health facility, awareness of the government geriatric health scheme.

## Discussion

This study identified the self-reported chronic morbidities, health problems and health care seeking and utilization behaviour among the elderly population residing in two sub-metropolitan cities of Province one, Nepal. These two cities are polarized in culture, ethnicity, religion, education, socioeconomic status, and lifestyle; therefore, the findings of this study represent the scenario of urban areas of the country. The availability of health care facilities in these two cities is also relatable to other urban areas of Nepal [31]. This study found a significant

**Table 6. Multivariable logistic regression analysis of different variables with health service utilization in the last months.**

| Characteristics | Category | AOR | 95% C.I. for Adjusted odds ratio (AOR) | | p-value |
|---|---|---|---|---|---|
| | | | Lower | Upper | |
| **Age group** | 80 years and above | | | | |
| | 60–69 years | 1.511 | 0.874 | 2.612 | 0.14 |
| | 70–79 years | 1.873 | 1.078 | 3.252 | 0.02 |
| **Economic status** | Below poverty line | | | | <0.01 |
| | Above poverty line | 2.286 | 1.538 | 3.397 | |
| **Pre-existing morbidities** | None | | | | <0.01 |
| | Single morbidity | 2.435 | 1.581 | 3.751 | |
| | Multimorbidity | 6.156 | 3.306 | 11.463 | |
| **Health Insurance** | No | | | | <0.01 |
| | Yes | 2.244 | 1.329 | 3.789 | |
| **Awareness of the government geriatric health scheme** | No | | | | <0.01 |
| | Yes | 2.553 | 1.713 | 3.805 | |

Variables adjusted with age, gender, occupation, economic status, ability to read and write, self-reported health status, preexisting comorbidities, health insurance, awareness of the government geriatric health scheme, living arrangements.

proportion of the elderly suffering from preexisting chronic morbidities. In the health problems, the highest prevalence was circulatory problems, and multiple health ailments were common among the elderly. This study found that 68.1% of the elderly people were facing difficulty in seeking health care and only 61.7% of the elderly have visited a health care facility within a year duration, only 35.5% had a regular visit to a doctor, and 8.9% reported never visited health facility till the study period.

## Chronic morbidities

This study found that almost half of the participants (48.3%) had pre-existing chronic morbidity at the time of the study, of which 30.9% had single morbidity and 17.4% had multimorbidity. A study done in eastern Nepal has reported a prevalence of preexisting morbidity to be 66.5%, of which 43.8% had single morbidity and 22.8% had multimorbidity [32]. Another study done in eastern Nepal has also found a significant burden of morbidity among the elderly which is higher than our study [33]. It may be the result of the inclusion of osteoarthritis as chronic morbidity in this study while our study has not considered osteoarthritis in preexisting morbidity.

Consistent with our study finding, a multinational study including low and middle-income countries has also discovered a similar burden of hypertension [34]. Expectedly, the findings of our study reflect the similar burden of hypertension and diabetes found by a study done in eastern Nepal among the elderly [33]. The finding of our study shows a slight increment in the prevalence of hypertension and diabetes but a decrease in cases of chronic obstructive pulmonary diseases (COPD) compared to the study done in Dharan municipality in 2007 which represents the increasing trend of hypertension and diabetes in the elderly community [35]. A study analyzing the worldwide trend of hypertension has concluded that the burden of hypertension among 30–79 years doubled from 1990 to 2019 [36]. The reduction in COPD may have resulted due to decreased use of biomass fuel with the introduction of clean energy [37]. The study from Bhaktapur, Nepal also reports a similar burden of hypertension, but a higher burden of COPD than our study [3]. This discrepancy may be due to the variation in sample size, geographical distribution, ethnicity and cultural practices. Other studies have reported a prevalence of hypertension ranging from 27–57% among the elderly [35, 38–45]. The prevalence of hypertension in the STEPS survey 2019 has shown a higher burden of hypertension among the increasing age group [46]. Like our study, a study done in Pune and Karnataka India reported the highest burden of hypertension followed by diabetes, heart disease and the lowest in asthma among the chronic morbidities in the elderly [39, 47]. A multinational study done in 29 low-middle-income countries has stated an overall prevalence to be 7.5% which is almost half of the findings of our study [48]. This may be due to the difference in sample population as this study has included participants of 25 years and above.

## Health problems among the elderly

Health problems like circulatory problems, digestive problems, musculoskeletal problems and psychological problems as reported in this study are the common problems of the ageing population [8]. Moreover, cardiovascular diseases are the leading cause of death globally [49]. The prevalence of eye problems reported by our study was higher than the findings of a study done in Bhaktapur but lower than the findings of studies done in Dharan and India [3, 35, 39]. Hearing impairment among the elderly in this study was complained by 29.8% akin to a study done in Dharan but different from a study done in India which reported a relatively higher prevalence [35, 39]. Dental problem reported by our study was relatively higher than that reported by a study done in India [39]. These discrepancies in findings may be the result of

tools used for identifying these health problems in different studies. Moreover, the questionnaires of our study were more focused on exploring the relative needs of the elderly but confirming the real need is the limitation of the study.

Consistent with our finding, a study done in Bhaktapur and Dharan has reported a similar burden of musculoskeletal problems among the elderly [3, 35]. A study in Nepal has found musculoskeletal problems to be the most common morbidity among the elderly [45]. Moreover, a study done in Nepal has also reported an association between joint pain and advancing age [50]. A multinational study has also stated low back pain to be the leading cause of years lived with disability in 2017 [1].

### Health seeking behaviour and health care utilization

The health service utilization, emergency visits and admission rate differ in our study from than findings of a study done in Butwal city in west Nepal, which may be due to differences in sample sizes, geographical differences and health facility availability. Regarding the utilization of the traditional healing system both the study reported a significant proportion of elderly visiting traditional healing which suggest a common belief and cultural alignments [51]. Similarly, a study done in the Ilam district of Nepal reported one-fifth of the population sought a traditional healer's service, supporting a strong belief in the traditional healing system [52]. In contrast to our findings, a study exploring health care utilization for headache disorders found the majority of visiting paramedical professionals for their problems [53]. This may be due to differences in accessibility of health services, health information, and the perceived credibility and trustworthiness of studied participants [54].

In our study gender, economic status, walking distance of health facility, self-reported health and perceived geriatric health policy had a major role in difficulty in seeking health care. Similarly, economic status, health insurance, preexisting chronic morbidities and perceived geriatric health policy constituted a major contributor influencing health care utilization among the elderly. Other studies of Nepal have also recognized the difference in household economic status, family income, chronic disease, educational status and self-rated health as significant determinants of health service utilization [52, 55]. A study done in eastern Nepal has shown a significant association between preexisting morbidity with difficulty in seeking health care and acquiring medication [32]. Another study in Nepal has found an increasing health care utilization proportional to symptom severity [53]. A study done in China has reported a crucial role of the need factor in determining health service utilization among the elderly. This same study has reported financial difficulties as a barrier and education, having social security and poor health status as a facilitator to use health services [56]. Similar to our finding, a study done in India among the elderly has found health care utilization to increase significantly with multimorbidity [57]. Moreover, studies from outside Nepal have also reported frailty multimorbidity and disability to impact doctor visits [58–60]. Similarly, a study done in Norway found medium- and high-risk patients more likely to healthcare utilization compared with low-risk patients [61]. An all of Us nationwide survey have reported financial concerns and lack of access to transportation as a reason for the delay in seeking care [62]. A study in North Carolina also stated several geographic and spatial behaviour factors, including having a driver's license, use of provided rides, and distance for regular care was significantly related to health care utilization [63].

### Conclusion

Our study unveiled a high burden of chronic morbidities and other health ailments among the elderly population as about half reported to have a preexisting chronic condition and almost

all reported having some sort of health problems. Despite substantial health needs, the participants did not adequately visit health care providers. The economic status and awareness of the government geriatric health scheme were the significant contributor to both health seeking behaviour and health care utilization among the elderly. Moreover, health insurance and pre-existing morbidities increased the odds of health care utilization. Continuous intervention and health education programs incorporating components like geriatric health policy, health insurance and other factors focusing on healthy ageing should be conducted to motivate elderly people for healthy behaviours, adequate health care seeking and utilization. The health policies should prioritise the ageing population and health services should be made available at the primary level with a geriatric specialized one-door policy capable of addressing common health problems among the elderly.

## Limitations

- This study tried to explore the common health problems among the elderly population yet missed one of the major issues of cognitive impairment.

- Mental health self-reports are sometimes subject to bias because of a general community stigma towards mental illness in Nepal [64].

## Supporting information

**S1 File.**
(DOCX)

**S2 File.**
(CSV)

## Acknowledgments

We are indebted to the participants of this study for providing us with their valuable time, information, help and cooperation.

## Author Contributions

**Conceptualization:** Mukesh Poudel, Asmita Ojha, Deepak Kumar Yadav.

**Data curation:** Mukesh Poudel, Jeevan Thapa.

**Formal analysis:** Jeevan Thapa.

**Investigation:** Mukesh Poudel.

**Methodology:** Mukesh Poudel, Asmita Ojha, Jeevan Thapa, Shyam Sundar Budhathoki.

**Supervision:** Deepak Kumar Yadav, Ram Bilakshan Sah, Avaniendra Chakravartty, Anup Ghimire, Shyam Sundar Budhathoki.

**Writing – original draft:** Mukesh Poudel.

**Writing – review & editing:** Mukesh Poudel, Asmita Ojha, Jeevan Thapa, Avaniendra Chakravartty, Shyam Sundar Budhathoki.

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
