## [Decision Letter · Decision Letter 0]

11 Nov 2021

PONE-D-21-29483Health needs, behavioural characteristics and health care utilization among elderly residing on urban areas of eastern Nepal. A cross sectional studyPLOS ONE

Dear Dr. Poudel,

Thank you for submitting your manuscript to PLOS ONE. After careful consideration, we feel that it has merit but does not fully meet PLOS ONE’s publication criteria as it currently stands. Therefore, we invite you to submit a revised version of the manuscript that addresses the points raised during the review process.

Kindly address the comments and concerns put forward by reviewers one and two. Also consider strengthening the statistical analysis by including a regression model as suggested by the reviewers. 

We look forward to receiving your revised manuscript.

Kind regards,

Pranil Man Singh Pradhan, M.D.

Academic Editor

PLOS ONE

4. We noted in your submission details that a portion of your manuscript may have been presented or published elsewhere. [DETAILS AS NEEDED] Please clarify whether this [conference proceeding or publication] was peer-reviewed and formally published. If this work was previously peer-reviewed and published, in the cover letter please provide the reason that this work does not constitute dual publication and should be included in the current manuscript.

Reviewers' comments:

Reviewer's Responses to Questions

**Comments to the Author**

1. Is the manuscript technically sound, and do the data support the conclusions?

Reviewer #1: Partly

Reviewer #2: Yes

2. Has the statistical analysis been performed appropriately and rigorously? 

Reviewer #1: No

Reviewer #2: Yes

3. Have the authors made all data underlying the findings in their manuscript fully available?

Reviewer #1: Yes

Reviewer #2: Yes

4. Is the manuscript presented in an intelligible fashion and written in standard English?

Reviewer #1: No

Reviewer #2: Yes

5. Review Comments to the Author

Reviewer #1: Dear authors,

Thank you for addressing an important issue concerning the older population in Nepal. Although the topic is important, it is not novel. It is not clear what is new in comparison to similar studies conducted in Nepal in the past. You may also consider conducting multivariate logistic/linear regression or other higher analysis (as appropriate) as this study only has descriptive data. Higher analyses will also make your study stronger. Please find my detailed comments below.

Line 57: What are the challenges and additional demands?

Lines 60-63: It is not clear what the unmet needs were and the purpose/objective of follow-up.

Line 96: Numbers less than ten should be spelled out in academic writing

Line 122: Check the spelling of “than” or “then”. “the complaints and findings were then….”

Line 166: Check spellings

Lines 165-169: Logistic regression is missing in the result section.

Proteinuria as a variable is not mentioned in the methods section.

Line 236: The term “developing country” may not be appropriate. The authors may consider using another term and also instead of using “like ours”, it is preferable to use “like/such as Nepal”.

Lines 235-239: The first paragraph of the discussion should mention the major findings of the study.

Discussion: Do not repeat sentences from the result section in discussion.

Discussion of major findings is lacking in general.

The conclusion section is weak and can be improved.

Academic writing:

1. Avoid use of unnecessary capitalizations

2. Use past tense to report methods and results

3. Use hyphenation where necessary (e.g.: face-to-face)

4. Inconsistent use of percentage (example: 53%, 27. 2%, 56.9% etc.)

5. Inconsistent use of the Oxford comma before “and”.

6. Commas missing in many sentences.

7. Check spellings

8. Inconsistent use of the words “aging” and “ageing”; “elderly” and “older people”. Use the same term throughout the text.

9. Recheck the text. The flow of paragraphs can be improved.

Hope the suggestions are useful. Best wishes.

Reviewer #2: � The topic is interesting and explores an important area of public health research. The article is good. However, there are some changes that need to made to make the article even better.

Overall Suggestions:

o There are minor errors in grammar/language. Please kindly proofread the article.

o Also some rearranging and adding of text may be necessary.

o Please make sure that the numbers in tables are correct. Details are in my comments below.

Title page:

o Line 10: Please use the correct spelling of ‘Hetauda’

Abstract:

o Line 50: Keywords not in alphabetical order

Materials and Methods:

o Line 86: What do you mean by structured interview? You have mentioned that the questionnaire itself is semi-structured. Do you mean face to face interview?

o Line 102: Please explain how this was a stratified proportionate random sampling? Did you have a sampling frame? If yes, please mention it.

o Line 134: The term elderly is defined here. As this study is about elderly people, it should be moved it up in the text so that reader can understand the context

o Line 165: Please specify which version of SPSS was used

o 167-169: Where are the results of multivariate analysis? Where is the Chi-square test used?

Results

o Line 182: In Table 1: for Age distribution, the percentage does not sum up to 100.0%. Please correct it.

o Line 184-186: Not clear what the author wants to say and may be confusing for the readers. Please re-write to bring clarity.

o Line 200-203: How did you assess the blood pressure at the time of visit? Did you measure the blood pressure yourselves? It is not mentioned in the methodology or tools used.

o Line 209: Measurement of proteinuria by the researcher and the tools used for this process is not mentioned in the methodology.

What you are trying to present as health needs is not clearly depicted in the results and discussion section.

References:

o Some of the references are not according to the journal guidelines. Please make required changes.

6. PLOS authors have the option to publish the peer review history of their article (what does this mean?). If published, this will include your full peer review and any attached files.

Reviewer #1: No

Reviewer #2: No

---

## [Author Response · Author response to Decision Letter 0]

22 Feb 2022

Dear Editor,

Warm Regards

Sub: Submission of the revised manuscript

Sir,

I would like to heartily thank you for reviewing our manuscript and allowing us to submit a revised manuscript. We have considered all the comments received for this revision. In addition, we have made some changes (rewording of some terms) to the title and objectives to ensure consistency of our focus and improve the storyline in the manuscript. We have extensively revised the results section and also added some new tables. The title now reads, Morbidities, health problems, health care seeking and utilization behaviour among elderly residing on urban areas of eastern Nepal: a cross-sectional study. 

We have also added an author who had contributed to the manuscript throughout but was not available to consent for authorship during the first submission. We now have an agreement from the author. We have completed the ‘Authorship Changes’ form and attached it with this revision. We hope you consider this change in line with PLOS guidelines. We appreciate the time and effort that you and the reviewers have dedicated to providing your valuable feedback on this manuscript. We are grateful to the reviewers for their insightful comments on our paper. We have also attached a point-by-point response to the reviewers’ comments and concerns as required. 

I will upload my study’s minimal underlying data and survey questionnaire used as a supporting file. I confirm that this work does not constitute dual publication. Some of the findings of this work can be found in a preprint available on: https://assets.researchsquare.com/files/rs-9433/v2/fc5b8b50-f3de-4f1e-a1b0-6042ee5da034.pdf?c=1631830880.

Dear reviewers, I would like to thank you for your time and effort towards improvement of the manuscript. Below are the response to the specific points raised by reviewers in this manuscript.

Reviewer 1:

Thank you for the constructive comments. 

R1 Comment 1: Line 57: What are the challenges and additional demands?

Response: The challenges are to respond to the increasing health care demand and needs of elderly with increasing age. This line now reads as, “Morbidity increases with age, thus health care systems need to continually update in order to adequately cater for these additional demands.” 54-55.

R1 Comment 2: Lines 60-63: It is not clear what the unmet needs were and the purpose/objective of follow-up.

Response: The unmet needs are defined as no health care consultation despite of persistent health needs. I have edited this line and it now reads as “Lower survival probabilities for individuals with unmet health care needs are reported. Moreover, unmet health needs conferred a higher risk of mortality, with excess risk ranging from 10% to 155%” in line 57 - 59. 

R1 Comment 3: Line 96: Numbers less than ten should be spelled out in academic writing Line 

Response: This has been edited and the numbers are written in text accordingly.

R1 Comment 4: 122: Check the spelling of “than” or “then”. “the complaints and findings were then….”

Response: This has been edited in line 139.

R1 Comment 5: Line 166: Check spellings 

Response: We have checked all the spellings accordingly.

R1 Comment 6: Lines 165-169: Logistic regression is missing in the result section. Proteinuria as a variable is not mentioned in the methods section.

Response: We have added new tables with regression. Proteinuria is removed from methods as we are not reporting it in results. 

R1 Comment 7: Line 236: The term “developing country” may not be appropriate. The authors may consider using another term and also instead of using “like ours”, it is preferable to use “like/such as Nepal”.

Response: We have edited the text accordingly and rephrased the paragraph.

R1 Comment 8: Lines 235-239: The first paragraph of the discussion should mention the major findings of the study. Discussion: Do not repeat sentences from the result section in discussion. Discussion of major findings is lacking in general. The conclusion section is weak and can be improved.

Response: We have edited the discussion and conclusion section as advised. 

R1 Comment 9: Academic writing:

1. Avoid use of unnecessary capitalizations

Response: This has been checked and edited throughout.

2. Use past tense to report methods and results

Response: This has been checked and edited throughout.

3. Use hyphenation where necessary (e.g.: face-to-face)

Response: This has been checked and edited throughout

4. Inconsistent use of percentage (example: 53%, 27. 2%, 56.9% etc.)

Response: This has been checked and edited throughout

5. Inconsistent use of the Oxford comma before “and”.

Response: This has been checked and edited throughout

6. Commas missing in many sentences.

Response: This has been checked and edited throughout

7. Check spellings

Response: This has been checked and edited throughout

8. Inconsistent use of the words “aging” and “ageing”; “elderly” and “older people”. Use the same term throughout the text.

Response: This has been checked and edited throughout

9. Recheck the text. The flow of paragraphs can be improved. Hope the suggestions are useful. 

Best wishes.

Response: This has been checked and edited throughout. Thank you so much.

Reviewer #2: 

Thank you for the constructive comments. 

R2 Comment 1: �The topic is interesting and explores an important area of public health research. The article is good. However, there are some changes that need to made to make the article even better.

Response: Thank you so much. We have revised this manuscript considering all the comments. 

R2 Comment 2: Overall Suggestions: There are minor errors in grammar/language. Please kindly proofread the article.

Response: This has been checked and edited throughout

R2 Comment 3: o Also some rearranging and adding of text may be necessary.

Response: This has been checked and edited throughout

R2 Comment 4: o Please make sure that the numbers in tables are correct. Details are in my comments below.

Response: This has been checked and edited throughout

R2 Comment 5: �Title page:

o Line 10: Please use the correct spelling of ‘Hetauda’

Response: We have corrected it now 

R2 Comment 6: Abstract:

o Line 50: Keywords not in alphabetical order

Response: We have now kept keywords in alphabetical order.

R2 Comment 7: �Materials and Methods:

o Line 86: What do you mean by structured interview? You have mentioned that the questionnaire itself is semi-structured. Do you mean face to face interview?

Response: Yes, we meant face to face interview with semi-strucured questionnaire. We have corrected that.

R2 Comment 8: 

o Line 102: Please explain how this was a stratified proportionate random sampling? Did you have a sampling frame? If yes, please mention it.

Response: The sampling frame was elderly people residing two sub-metropolitan cities. Eight wards from each sub-metropolitan selected randomly were strata in this study. The population of wards were identified from administrative office and number of participants from each ward was calculated proportionately.

We have mentioned it in line 99 – 104.

R2 Comment 8: o Line 134: The term elderly is defined here. As this study is about elderly people, it should be moved it up in the text so that reader can understand the context 

Response: I have moved the text above in the introduction part and it is now in line 66 – 67. 

We have ensured this in the revision. 

R2 Comment 9: o Line 165: Please specify which version of SPSS was used

Response: We have specified the version of SPSS used. I used SPSS version 16. It is stated in line 159 -160. 

R2 Comment 10: o 167-169: Where are the results of multivariate analysis? Where is the Chi-square test used?

Response: We have added the tables with the results for chi-square test and multivariate analysis. Table 4 shows the bivariate analysis and table 5 and 6 states the finding of multivariable analysis.

R2 Comment 11: �Results

o Line 182: In Table 1: for Age distribution, the percentage does not sum up to 100.0%. Please correct it.

Response: This has been checked and corrected. Thank you.

R2 Comment 12: o Line 184-186: Not clear what the author wants to say and may be confusing for the readers. Please re-write to bring clarity.

Response: We have edited this to two categories, with family or by themselves. 

R2 Comment 13: o Line 200-203: How did you assess the blood pressure at the time of visit? Did you measure the blood pressure yourselves? It is not mentioned in the methodology or tools used.

Response: As a qualified physician, the PI measured the blood pressure. This is now mentioned in methodology and is in line 129 – 133.

R2 Comment 14: o Line 209: Measurement of proteinuria by the researcher and the tools used for this process is not mentioned in the methodology.

Response: Based on this revision after adding and rearranging the findings to match the objectives, we have removed proteinuria from results hence not mentioned it in methods. 

R2 Comment 15: �What you are trying to present as health needs is not clearly depicted in the results and discussion section.

Response: We have revised the title as ‘ Morbidities, health problems, health care seeking and utilization behaviour among elderly residing on urban areas of eastern Nepal: a cross-sectional study’ to improve the focus and enhance clarity. We have ensured this is reflected in the objectives as well as presented in a similar manner in results and discussion, 

R2 Comment 16: �References: Some of the references are not according to the journal guidelines. Please make required changes.

Response: We have made these changes in the references.

Thank you ever so much for all the comments.

---

## [Decision Letter · Decision Letter 1]

13 Jun 2022

PONE-D-21-29483R1Morbidities, health problems, health care seeking and utilization behaviour among elderly residing on urban areas of eastern Nepal: a cross-sectional studyPLOS ONE

Dear Dr. Poudel,

Thank you for submitting your manuscript to PLOS ONE. After careful consideration, we feel that it has merit but does not fully meet PLOS ONE’s publication criteria as it currently stands. Therefore, we invite you to submit a revised version of the manuscript that addresses the points raised during the review process.

We look forward to receiving your revised manuscript.

Kind regards,

Pranil Man Singh Pradhan, M.D.

Academic Editor

PLOS ONE

Journal Requirements:

Additional Editor Comments (if provided):

Please address the minor comments provided by the reviewers.

Reviewers' comments:

Reviewer's Responses to Questions

**Comments to the Author**

1. If the authors have adequately addressed your comments raised in a previous round of review and you feel that this manuscript is now acceptable for publication, you may indicate that here to bypass the “Comments to the Author” section, enter your conflict of interest statement in the “Confidential to Editor” section, and submit your "Accept" recommendation.

Reviewer #2: All comments have been addressed

Reviewer #3: (No Response)

2. Is the manuscript technically sound, and do the data support the conclusions?

Reviewer #2: Yes

Reviewer #3: Partly

3. Has the statistical analysis been performed appropriately and rigorously? 

Reviewer #2: Yes

Reviewer #3: I Don't Know

4. Have the authors made all data underlying the findings in their manuscript fully available?

Reviewer #2: Yes

Reviewer #3: Yes

5. Is the manuscript presented in an intelligible fashion and written in standard English?

Reviewer #2: Yes

Reviewer #3: Yes

6. Review Comments to the Author

Reviewer #2: The authors have done a good job in addressing all the comments. After the revision, the quality of the article has been enhanced.

Reviewer #3: Dear Authors,

The article is very well written and the enormous effort undertaken during the study design, data analysis and writing of the manuscript is greatly appreciated. Indeed, the issues of elderly is an important issue and in developing countries like Nepal, more such research and articles are appreciated.

There are few comments which might help in making the manuscript better.

General Writing: The article is well written but at some places, reorganization of the writing for a better flow can be considered (details in ‘review comment’ section). Minor grammar/punctuation/scientific writing errors are there like:

Line 26 (abstract) inconsistent use of hyphenation in ‘health care seeking’

Line 216 ‘6’ to be written as six

Line 276 comma missing.

Yes, the manuscript is well written. However, the data shared is coded and codes for limited variables are given so reviewer is unable to decide with full confidence. Eg GD column in data sheet reads as ‘multimorbidity’ which has 271 rows coded as 1 (51.13%). Reviewer cannot be sure what this implies. Most of the conclusions seem possible when the questionnaire is reviewed.

Line 99 Sampling method has been explained and eight wards have been selected as per the manuscript but data sheet shows missing data in 45 participants and there are 9 wards in one of the cities i.e. Dharan (if the number allotted after Dharan is the ward no in the data sheet). More clarity would be there if authors showed the number in sampling frame and ward wise population as reviewer believes this is a great strength of this article.

Line 112-116 Calculation of sample size: The formula used one proportion sample size seems fine, but in the stated formula, I am not sure where the power (85%) is used. The rationale for choosing prevalence of unmet health care needs in people suffering with ‘hypertension’ along with the full calculation would clarify readers.

Line 118 onwards in “variables studied’’ there is inconsistent writing style with some of the variables being described in paragraph and some with heading.

Line 119 The questionnaire is ‘pre-tested’. Elaborating the pre-testing technique would strengthen the article as the questionnaire shared is in great detail with translation to local language. This work is not reflected in the manuscript.

Line 134-136 BDI II scale used for depression but the result of this has not been expressed in ‘Results’ section. If authors are not publishing this, maybe they can omit from methods section too. It also raises a question why only ‘depression’ was assessed and not other mental health issues.

Line 140-141 ICPC is used for health problems which is a standard in itself so the sentence following it with Ref 30 seems only to add to the authors long list of references (study among inmates vs study of elderly)

Line 149 Reference for this variable (if provided) would be good.

Table 3

Mentions “Pention camp’’, the terminology might not be clear to many people so elaboration below table on what kind of ‘health facility’ would add to clarity.

In ‘Last visit to any health facility’, one of the options reads as ‘Never’ with 47 responders choosing it. Does it mean in their 60 years, they never visited a health facility?

Discussion Section:

Overall slightly weak. Authors have compared their findings with other researches but they have not explained why the variations are there. Eg for HTN, eye (66.2 vs 19.1), dental issues etc

Line 257 The article including the title of the article refers the study setting as urban eastern Nepal, while in this line the article claims to represent whole Nepal in terms of culture, ethnicity, religion, education, socioeconomic status, lifestyle which seems contradictory in itself.

Flow of article in chronic morbidities section needs to be worked out. Eg

Line 271-275 Manuscript is describing articles

Line 276-277 Manuscript gives a broad comment

Line 277 onwards manuscript starts to describe individual articles

When discussing, it is better to discuss one variable and then start others, broad statements are better at beginning of discussion to start a variable discussion or the end.

Line 277 The prevalence of HTN is 34% in this study. The manuscript reports a ‘similar’ range of 27-57% in other studies. Maybe the authors should reconsider the term ‘similar’ considering the expressed range.

References in discussion section is missing. Eg Line 280, Line 291, Line 303. It gives more clarity to readers if reference is given immediately after a particular article is discussed and not at the end of the paragraph. Author is advised to check this throughout the discussion section.

References:

One Ref 37 is of the year 2022 while most others are relatively not recent so maybe the author can add more recent articles in discussion section while revising the manuscript. Author has 56 references; some references might need reconsideration for citation as stated above.

7. PLOS authors have the option to publish the peer review history of their article (what does this mean?). If published, this will include your full peer review and any attached files.

Reviewer #2: No

Reviewer #3: No

---

## [Author Response · Author response to Decision Letter 1]

23 Jul 2022

Dear reviewer,

I kindly appreciate all your constructive comments and effort in making this manuscript better. I have tried my best to respond to all your comments.

Reviewer 2

Thank you for the appreciation.

Reviewer 3

Thank you for the constructive comments.

R3 Comment 1: The article is very well written and the enormous effort undertaken during the study design, data analysis and writing of the manuscript is greatly appreciated. Indeed, the issues of elderly is an important issue and in developing countries like Nepal, more such research and articles are appreciated.

Response: Thank you for the appreciation. We will continuously look forward to explore the issues of elderly and are motivated to conduct research in this area

R3 Comment 2: Line 26 (abstract) inconsistent use of hyphenation in ‘health care seeking’

Response: I have removed the hyphenation and the word now reads as health care seeking behaviour in line 26.

R3 Comment 3: Line 216 ‘6’ to be written as six

Response: This has been edited and number is written in text in line 221.

R3 Comment 4: Line 276 comma missing.

Response: This has been edited.

R3 Comment 5: Yes, the manuscript is well written. However, the data shared is coded and codes for limited variables are given so reviewer is unable to decide with full confidence. Eg GD column in data sheet reads as ‘multimorbidity’ which has 271 rows coded as 1 (51.13%). Reviewer cannot be sure what this implies. Most of the conclusions seem possible when the questionnaire is reviewed.

Response: Thank you. We shared a basic excel sheet and a SPSS data set. The excel sheet being a basic data set we tried various outcomes. The multimorbidity stated in that data set includes the relative health problem also. But later for the analysis the morbidity and multimorbidity was classified based upon preexisting health condition. The SPSS data set has been recoded again for analysis and it gives more clarity on data analysis. 

R3 Comment 6: Line 99 Sampling method has been explained and eight wards have been selected as per the manuscript but data sheet shows missing data in 45 participants and there are 9 wards in one of the cities i.e. Dharan (if the number allotted after Dharan is the ward no in the data sheet). More clarity would be there if authors showed the number in sampling frame and ward wise population as reviewer believes this is a great strength of this article.

Response: The ward wise population has been cited accordingly in the methodology section in line 104. The data representing ward number has been revised and corrected accordingly.

R3 Comment 7: Line 112-116 Calculation of sample size: The formula used one proportion sample size seems fine, but in the stated formula, I am not sure where the power (85%) is used. The rationale for choosing prevalence of unmet health care needs in people suffering with ‘hypertension’ along with the full calculation would clarify readers.

Response: The calculation of sample size is elaborated accordingly in line 112-118. The rationale of choosing prevalence of unmet health care need is now stated which reads as “This prevalence suggests the scenario of health care seeking practice with one of the common morbidity among elderly” in line 114-115.

R3 Comment 8: Line 118 onwards in “variables studied’’ there is inconsistent writing style with some of the variables being described in paragraph and some with heading.

Response: This has been edited and variables are now described in paragraph.

R3 Comment 9: Line 119 The questionnaire is ‘pre-tested’. Elaborating the pre-testing technique would strengthen the article as the questionnaire shared is in great detail with translation to local language. This work is not reflected in the manuscript.

Response: The pretesting was done among the elderly visiting the BP Koirala Institute of Health Sciences. We have added about pretesting in the manuscript as “A questionnaire intended to fulfil the study objective was developed based on different studies among the elderly. The questions were discussed among authors for validity. The translation of questions into the local language was done. Elderly visiting the BP Koirala Institute of Health Sciences were interviewed. Authors reviewed the acquired answers with an amendment to the questionnaire.” In line 122-126.

R3 Comment 10: Line 134-136 BDI II scale used for depression but the result of this has not been expressed in ‘Results’ section. If authors are not publishing this, maybe they can omit from methods section too. It also raises a question why only ‘depression’ was assessed and not other mental health issues.

Response: It has been presented in result section as psychological problem. The rationale of assessing this mental health problem is now included and it reads as “Depression is one of the most common psychological problem in elderly population” in line 142-143.

R3 Comment 11: Line 140-141 ICPC is used for health problems which is a standard in itself so the sentence following it with Ref 30 seems only to add to the authors long list of references (study among inmates vs study of elderly)

Response: By citing the paper on inmates, our intentions were to show that ICPC has been used for research in Nepal. However to avoid confusion the reference has been revised and removed in line 148.

R3 Comment 12: Line 149 Reference for this variable (if provided) would be good.

Response: The reference for tobacco users has now been provided to that variable in line 156.

R3 Comment 13: Table 3 Mentions “Pention camp’’, the terminology might not be clear to many people so elaboration below table on what kind of ‘health facility’ would add to clarity.

Response: The footnote in the table now describes in a brief about the pension camp which reads as “Pension camp is the common term used by people for pension paying office for the Ex Gurkha army personnel, which also have a health facility with medical personnel and doctors which provides health care service free of cost” in line 216 – 218.

R3 Comment 13: In ‘Last visit to any health facility’, one of the options reads as ‘Never’ with 47 responders choosing it. Does it mean in their 60 years, they never visited a health facility?

Response: Yes, it was the response from those 47 participants. The added response of these participant was taking home remedies and seeking traditional healer.

R3 Comment 14: Discussion Section: Overall slightly weak. Authors have compared their findings with other researches but they have not explained why the variations are there. Eg for HTN, eye (66.2 vs 19.1), dental issues etc

Response: We have revised the discussion and added explanation for such variations in line 312 -315.

R3 Comment 15: Line 257 The article including the title of the article refers the study setting as urban eastern Nepal, while in this line the article claims to represent whole Nepal in terms of culture, ethnicity, religion, education, socioeconomic status, lifestyle which seems contradictory in itself.

Response: We have revised it and edited as “represent urban areas of the country” in line 264. 

R3 Comment 16: Flow of article in chronic morbidities section needs to be worked out. Eg

- Line 271-275 Manuscript is describing articles 

- Line 276-277 Manuscript gives a broad comment

- Line 277 onwards manuscript starts to describe individual articles

- When discussing, it is better to discuss one variable and then start others, broad statements are better at beginning of discussion to start a variable discussion or the end.

Response: We have revised the discussion and edited as per the comments in line 273-301.

R3 Comment 17: Line 277 The prevalence of HTN is 34% in this study. The manuscript reports a ‘similar’ range of 27-57% in other studies. Maybe the authors should reconsider the term ‘similar’ considering the expressed range.

Response: This has been revised and edited. Similarly has been removed and it has been stated as “Other studies have reported a prevalence of hypertension ranging from 27-57% among the elderly” in line 293 - 294.

R3 Comment 18: References in discussion section is missing. Eg Line 280, Line 291, Line 303. It gives more clarity to readers if reference is given immediately after a particular article is discussed and not at the end of the paragraph. Author is advised to check this throughout the discussion section.

Response: We have revised this and edited accordingly. The references is now provided after each statement.

R3 Comment 19: References:

One Ref 37 is of the year 2022 while most others are relatively not recent so maybe the author can add more recent articles in discussion section while revising the manuscript. Author has 56 references; some references might need reconsideration for citation as stated above.

Response: This has been revised and we have tried to include the recent articles in discussion section.

---

## [Editor Report · Decision Letter 2]

3 Aug 2022

Morbidities, health problems, health care seeking and utilization behaviour among elderly residing on urban areas of eastern Nepal: a cross-sectional study

PONE-D-21-29483R2

Dear Dr. Poudel,

We’re pleased to inform you that your manuscript has been judged scientifically suitable for publication and will be formally accepted for publication once it meets all outstanding technical requirements.

Kind regards,

Pranil Man Singh Pradhan, M.D.

Academic Editor

PLOS ONE
---

## [Editor Report · Acceptance letter]

26 Aug 2022

PONE-D-21-29483R2 

Morbidities, health problems, health care seeking and utilization behaviour among elderly residing on urban areas of eastern Nepal: a cross-sectional study 

Dear Dr. Poudel:

I'm pleased to inform you that your manuscript has been deemed suitable for publication in PLOS ONE. Congratulations! Your manuscript is now with our production department. 

Kind regards, 

on behalf of

Dr. Pranil Man Singh Pradhan 

Academic Editor

PLOS ONE